# Advances in Vitreoretinal Surgery

**DOI:** 10.3390/jcm11216428

**Published:** 2022-10-30

**Authors:** Lucas Ribeiro, Juliana Oliveira, Dante Kuroiwa, Mohamed Kolko, Rodrigo Fernandes, Octaviano Junior, Nilva Moraes, Huber Vasconcelos, Talita Oliveira, Mauricio Maia

**Affiliations:** Department of Ophthalmology, Federal University of São Paulo, São Paulo 04021-001, Brazil

**Keywords:** retinal detachment, vitreoretinal surgery, vitrectomy, vitrectomy instrumentation, macular holes

## Abstract

Advances in vitreoretinal surgery provide greater safety, efficacy, and reliability in the management of the several vitreoretinal diseases that benefit from surgical treatment. The advances are divided into the following topics: scleral buckling using chandelier illumination guided by non-contact visualization systems; sclerotomy/valved trocar diameters; posterior vitrectomy systems and ergonomic vitrectomy probes; chromovitrectomy; vitreous substitutes; intraoperative visualization systems including three-dimensional technology, systems for intraoperative optical coherence tomography, new instrumentation in vitreoretinal surgery, anti-VEGF injection before vitrectomy and in eyes with proliferative diabetic retinopathy, and new surgical techniques; endoscopic surgery; the management of subretinal hemorrhages; gene therapy; alternative techniques for refractory macular hole; perspectives for stem cell therapy and the prevention of proliferative vitreoretinopathy; and, finally, the Port Delivery System. The main objective of this review is to update the reader on the latest changes in vitreoretinal surgery and to provide an understanding of how each has impacted the improvement of surgical outcomes.

## 1. Introduction

The understanding of vitreoretinal diseases has evolved with the development of new diagnostic technologies; similarly, the indications for surgical approaches have evolved in parallel. For this reason, new surgical techniques and technologies have contributed to the advances of vitreoretinal surgery since Robert Machemer performed a vitreoretinal surgery in 1970. Initially considered an unsafe, ineffective surgery, it has become highly technologic, safe, and predictable in most cases [1].

Vitrectomy has been transformed from 17-gauge (G) systems and 400 cuts/minute (cpm) to 27-G sclerotomy systems and even 20,000 cpm, making it a more effective and less traumatic surgery in selected cases [2]. In addition, advances include a better control of intraocular pressure during eye surgery and an improved quality of intraoperative visualization.

## 2. Development

Vitreoretinal surgery has undergone the most changes in recent decades. New technologies require that possible advantages be scientifically proven to be considered “state-of-the-art”, and there are limitations for rapid adoption, such as excessive cost, learning curve, and surgeon preference for the techniques previously considered the standard.

Advances in vitreoretinal surgery can be divided, based on a didactic point of view, into 13 fields: scleral buckling using the chandelier illumination guided by non-contact visualization systems; sclerotomy/valved trocar diameters; posterior vitrectomy systems and ergonomic vitrectomy probes; chromovitrectomy; vitreous substitutes; intraoperative visualization systems including three-dimensional (3D) technology and systems for intraoperative optical coherence tomography (OCTi); new instrumentation in vitreoretinal surgery; anti-VEGF injections before vitrectomy in eyes with proliferative diabetic retinopathy (PDR); and new surgical techniques such as endoscopic surgery, the management of subretinal hemorrhages, gene therapy using viral vectors, alternative techniques for refractory macular holes (MHs), and perspectives for stem cell therapy and retinal pigment epithelium (RPE transplantation); the prevention of proliferative vitreoretinopathy (PVR); and the Port Delivery System.

### 2.1. Advances in Management of Rhegmatogenous Retinal Detachments

#### 2.1.1. Primary Vitrectomy

Rhegmatogenous retinal detachments (RRDs) can be treated with different surgical procedures, and controversy remains about the preferable operating method of more complex RDs not complicated by PVR. The proportion of aphakic/pseudophakic patients with RRD has increased to 30% during the past decade due to the increasing numbers of performed cataract operations. Primary pars plana vitrectomy (PPV) (Figure 1) has recently gained popularity for treating RRDs, especially in pseudophakic eyes [3].

A study that compared PPV and scleral buckling techniques in patients with RRDs uncomplicated by PVR reported better anatomic outcomes in pseudophakic eyes [4].

#### 2.1.2. Scleral Buckling Using the Chandelier Illumination Guided by Non-Contact Visualization Systems

Since its inception in the 1950s and subsequent decades, scleral buckling has been a useful surgical technique to repair RRDs, especially in young, phakic patients and in the presence of inferior breaks [4]. Traditionally, retinal visualization during scleral buckling relies on indirect ophthalmoscopy to localize and apply cryotherapy to the breaks, confirm adequate subretinal fluid drainage, and adjust the buckle height, where the view may be hampered by poor pupillary dilation and media opacities.

Another technique used for primary rhegmatogenous retinal detachments is scleral buckling using a surgical microscope. The core technique is the use of a surgical microscope to perform scleral buckling surgery, directly visualize retinal breaks or lesions, and then perform transscleral cryopexy without using any contact lenses. To perform this technique, it is necessary to perform a pre-operative examination with a three-mirror contact lens to locate the areas of the retinal breaks and degeneration and to intraoperatively guide the scleral buckling and cryotherapy of retinal lesions [5,6]. This technique has the following advantages: it uses an erect image, easily enables accurate sutures, and allows the pre-equatorial and peripheral retina to be easily seen. However, this technique has a restricted view, it is not possible to evaluate the posterior pole, and it needs to put strong pressure on the peripheral retina to access the areas of rupture [5].

Recent advances in visualization use a surgical microscope and a 25-G and 27-G cannula-based endoillumination system, also known as a chandelier lighting system [7,8] (Figure 2). Twenty-seven-gauge endoilluminator-assisted scleral buckling is an easy and safe procedure that provides a better control over and free adjustment of the light direction, thus overcoming the limitations of chandelier-assisted surgery [8,9].

Aras and Ucar first described transscleral fiber optic-assisted scleral buckling to repair RDs in 2012 [10]. This initial method used a torpedo-style chandelier light source through an uncannulated sclerotomy associated with a non-contact wide-angle viewing system to allow for the better identification and treatment of all retinal breaks.

The potential advantages of chandelier-assisted buckling include the improved visualization of the peripheral retina, direct viewing during the external drainage of subretinal fluid, the ability to convert to PPV, enhanced teaching capabilities, and improved ergonomics [11]. The limitations include the need for additional instruments such as a wide-angle viewing system and a chandelier illumination system, making it more expensive and instrument-dependent [12].

Despite many benefits, chandelier endoillumination-assisted scleral buckling has not been widely accepted in clinical practice. The main reason is that conventional scleral buckling is an extraocular procedure, but chandelier endoillumination must be inserted into the vitreous cavity, which may increase potential risks such as infectious endophthalmitis, lens damage, vitreous incarceration or tissue proliferation, and phototoxicity [13].

### 2.2. Sclerotomy/Valved Trocar Diameters

Vitrectomy with 20-G sclerotomy, which maintained state-of-the-art status for over 30 years, lost popularity in the beginning of the 21st century when 23-G vitrectomy was introduced. The newer system caused less tissue damage and had a lower risk of iatrogenic retinal breaks, shorter operating time, and better conditions for surgical incision closure while maintaining a similar efficiency [14].

The emergence of 25-G vitrectomy in 2003 did not immediately replace 23-G vitrectomy because challenges arose due to the smaller caliber of instrumentation and intraoperatively reduced cut and flow rates, making it inefficient for some surgeons. The system only proved to be effective in selected cases of stage 5 retinopathy of prematurity, which required a more meticulous approach because of the lower aspiration rate [15]. As such, 25-G vitrectomy did not become popular for most vitreoretinal diseases. With advancements in the development of equipment and instruments, however, 23-G and 25-G vitrectomy became the main choices for most surgeons [16].

The smallest-diameter sclerotomy for vitrectomy is currently 27-G, which still faces the same challenges as the 25-G system. Microincision sutureless vitrectomy (MISV) instruments have provided numerous advantages, including shorter operative times, self-sealing scleral wounds, decreased postoperative pain and inflammation, decreased astigmatism, and faster visual recovery [17]. Nevertheless, they encounter problems during stages that require higher flow and cutting speed, especially in the vitreous core [17]. The lighting provided by the smaller instruments also is reduced.

### 2.3. Posterior Vitrectomy Systems and Ergonomic Vitrectomy Probes

Posterior vitrectomy devices have markedly improved since the arrival of the first models, with important advancements in safety and surgical effectiveness. The advances have also allowed for important improvements in efficiency for 27-G [17] systems. The three main currently available platforms are the Constellation (Alcon, Fort Worth, TX, USA), Stellaris Elite (Bausch + Lomb, Rochester, NY, USA), and EVA (DORC, Zuidland, The Netherlands).

The Constellation includes an advanced Hypervit probe with double cutting and improved aspiration, cut speed, and flow compared with the previously used Ultravit tip [18]. The technology creates about 25% to 41% less traction during 27-G vitrectomy at 20,000 cpm compared with the Ultravit with 10,000 cpm [19]. The lighting provided by the chandelier is markedly more effective and has become indispensable for 27-G surgeries.

The Stellaris Elite includes the new Bi-Blade tip, which facilitates faster vitreous aspiration, especially during 25-G and 27-G surgeries. In addition, the Vitesse hypersonic vitrectomy pen, which liquefies the vitreous instead of using traditional cutting systems, results in less vitreous traction in the periphery.

DORC’s EVA also uses a double-cut tip and has improved aspiration and flow rates during 25-G and 27-G vitrectomies.

All systems are effective and have advantages and disadvantages, allowing surgeons to choose their preferences.

### 2.4. Chromovitrectomy

Chromovitrectomy is defined as the use of vital dyes for the better visualization of tissue during vitrectomy. The vitreous is intraoperatively removed and replaced by a balanced saline solution. The removal of the posterior vitreous, called the posterior hyaloid, is a crucial step in vitrectomy, in that it is challenging to correctly identify the structures without using dyes. Dyes also are helpful during the removal of the internal limiting membrane (ILM), the interface between the posterior hyaloid and the retina that is only 10 µm thick [20].

A failure to properly visualize structures can cause inaccurate movements and irreversible damage to structures that are fundamental to vision. Chromovitrectomy was developed about two decades ago to minimize the risks during the stages [21].

Indocyanine green (ICG) has been used since 1970 for angiographic studies of the retina and choroid. The first studies of its use as a dye in vitrectomy, published in the 2000s, reported its affinity for the ILM [22]. Even though it has become one of the most used dyes for ILM peeling, there is potential for toxicity to the RPE and retina, which is why it should be used for the shortest possible time, in low doses, and with less light exposure [23].

Infracyanine green is similar to ICG, with the main difference being the absence of iodine in its composition, a factor that seems to be related to toxicity in the RPE. As a result, it is considered safer than ICG for ILM peeling but is more expensive [24].

Brilliant blue has emerged as an alternative dye for ILM peeling (Figure 3), with low toxicity to the RPE and retina. It has granular characteristics that facilitate handling and dilution in intraocular irrigations, making it easier to use compared with other dyes for ILM peeling [25]. In addition, unlike ICG, it does not need fluid–air exchange to increase its effectiveness, and because it is not fluorescent, it does not carry an increase in the risk of phototoxicity [26].

Trypan blue was initially used in 2000 to dye the anterior capsule during cataract surgery and donor endothelial cells in endothelial corneal transplantation, and it was later for evaluated chromovitrectomy [27]. Trypan blue has an affinity for tissues with high cellular proliferation, such as epiretinal membranes (ERMs), and it is less effective for staining the ILM compared with ICG [21]. Toxicity is minimal when used in low concentrations.

Patent blue has similar characteristics to those of trypan blue, and it was approved for use in ophthalmology in 2003 to stain the anterior lens capsule during cataract surgery and chromovitrectomy. Like trypan blue, it has a low toxicity to the RPE and higher affinity for glial tissues in ERM cases [28].

Membrane-Blue-Dual (MBD) consists of a combination of trypan blue (0.15%), brilliant blue G (0.025%), and 4% polyethylene glycol (PEG). This dye stains the ILM as well as other membranes (ERMs and proliferative vitreoretinopathy membranes), and it does not require an air–fluid exchange prior to dye injection because of its heavier molecular weight [29].

Triamcinolone is a water-insoluble synthetic corticosteroid that has been used in ophthalmology since 1980 and in chromovitrectomy since 2000 [30]. It is an excellent vitreous dye, as it is crystal in form and whitish in color, which provides contrast for the visualization of areas of interest (Figure 4). It is the most commonly used substance to identify the vitreous and has the potential advantages of reducing blood–ocular barrier breakage and the need for re-operations [31].

In summary, the choice of dyes should consider the type of surgery performed and the surgeon’s personal preference. The substances mostly used to stain ILMs are ICG, infracyanine green, and brilliant blue. Those used most often for posterior hyaloid and ERMs are trypan blue, patent blue, and triamcinolone.

Among recent advances in chromovitrectomy, two Brazilian studies stand out. One study evaluated the addition of lutein–zeaxanthin crystals to brilliant blue to protect the photoreceptors and RPE; however, this is seldom used in clinical practice [32]. The second study evaluated the dye extracted from açaí (*Euterpe oleracea*), which was developed in 2013 and is still in phase I/II clinical trials. This dye is effective for staining ILMs and posterior hyaloids [33] (Figure 5).

### 2.5. Vitreous Substitutes

In recent years, the development of vitreoretinal surgery has led to improvements of the vitreous substitutes. The vitreous is a complex, gelatinous structure that has important biomechanical, optical and physiological functions. None of the available substances are ideal vitreous substitutes, mainly serving as temporary or permanent retinal tamponade. There are two categories of substitute: gas-based and liquid [34].

Air is colorless, inert, inexpensive, and easy to find. However, it is easily absorbed by red blood cells, reducing its tamponade effect in a few days [35]. Therefore, its use is limited as a retinal tamponade, but it allows for the faster recovery of vision [34].

Other tamponade gases have been important for vitreoretinal surgery since the 1970s. Today, sulfur hexafluoride (SF_6_) and perfluoropropane (C_3_F_8_) are increasingly being used in the treatment of many complicated vitreoretinal diseases [34]. Both these gases are heavier than air, colorless, odorless, and nontoxic [35]. Sulfur hexafluoride expands to double the injected volume within 1 to 2 days and lasts in the vitreous cavity for 1 to 2 weeks. Perfluoropropane expands to about four times its original volume in 72 to 96 h and lasts for 6 to 8 weeks [35]. Adverse effects include an increase in IOP during surgery and, for a few days after injection, gas-induced cataract formation and corneal endothelial changes.

The balanced saline solution (BSS) is an example of a liquid substitute, presenting physical characteristics very similar to the aqueous humor regarding transparency, refractive index, and density [36]. Saline solutions are used as temporary vitreous substitutes during exchange with air or liquids as they represent a simple filling liquid, with no tamponade properties on the retina due to its low surface tension [37].

Silicone oil (SO) is a is a hydrophobic polymer with a specific gravity slightly less than water (0.97 g/mL) and a refractive index similar to that of the vitreous [38]. All SO polymers are of commercial interest for their stability, lubricating properties, and as a vitreous substitute, with a high surface tension and viscosity, ease of removal, low toxicity, and transparency [34]. They are usually used for complicated retinal detachment, when postoperative airplane travel is planned, and in uncooperative patients. SO is available in several viscosities, but 1000 and 5000 centistokes are clinically used. SO is usually removed after 3 to 6 months to avoid complications such as cataract induction, corneal toxicity, glaucoma, and so-called “silicone retinopathy” [39].

### 2.6. Intraoperative Visualization Systems including 3D Technology

Proper visualization is essential to ensure an effective and safe surgery. Recent major changes in visualization systems have led to advances in surgery quality and comfort.

Previously, major milestones in intraoperative visualization were related to the type of systems used. Initially, contact systems allowed for a restricted visual range of up to 35 degrees and had severe limitations regarding visualization quality [31].

Wide-angle systems spurred a revolution in vitreoretinal surgery and were only made possible by the emergence of the Stereoscopic Diagonal Inverter (ADD MANUFACTURER, LOCATION) in 1987, which is an image inverter adapted for the optical structure of a microscope. The contact system, developed in 1989, requires direct contact between the lens and cornea and provides a good visualization of up to 130 degrees. The limitations include difficulties with scleral depression and the need for auxiliary help to intraoperatively hold the lens. The non-contact system, developed in 1987 (OCULUS BIOM, Port St. Lucie, FL, USA), provides easier handling and is widely accepted by surgeons [40].

The 3D heads-up visualization system was the last major revolution in visualization during vitreoretinal surgery. It was launched in 2016 [41] and is currently available on the Ngenuity (Alcon) and Artevo 800 (Carl Zeiss, Oberkochen, Germany) (Figure 6). The system uses a camera coupled to a microscope’s optics, which transmits the image to a television with a 3D system, requires the use of glasses, and eliminates the need for binoculars.

The main advantages of the new system are related to surgical teaching in that it allows an advisor to view the same image as a surgeon and provides ergonomics/surgeon comfort. Studies have not reported better surgical results compared with a traditional microscope [42,43]. Importantly, the 3D heads-up system does not exclude the need for a contact or non-contact visualization system.

### 2.7. Intraoperative OCT

The advent of OCT, which prompted a revolution in the diagnosis and understanding of vitreoretinal diseases, has become vital in the preoperative planning and postoperative follow-up of vitreoretinal surgeries, especially in pathologies involving the macular region [44].

The first attempts using OCT technology during a surgical procedure were in 2005, with time-domain OCT, and evaluated lamellar corneal transplants and trabeculectomy [45].

The development of portable spectral-domain OCT in 2009 enabled its use in vitreoretinal surgery through the following systems: EnVisu (handheld probe) (Leica, Wetzlar, Germany) and iVue (stand-mounted) (Optovue, Fremont, CA, USA). However, the images were of low quality because of artifacts, long learning curves, and difficulties in stabilization. In addition, it was not a real-time system and had a delay in the availability of images [46].

Image capture was facilitated using the EnVisu system coupled to a surgical microscope. This enabled the first multicenter study of intraoperative OCT (OCTi), PIONEER, in which 43% of surgeons reported that the system provided valuable information during the peeling of the ILM [47]. However, it was not a real-time system, and there was a delay between the capturing and actual viewing of the images.

Improvements in the quality and speed of image capture/processing occurred with the emergence of systems that were integrated into a surgical microscope, the commercially available Rescan 700 (Carl Zeiss), Enfocus (Leica), and OPMedT (Haag-Streit, Bern, Switzerland) (Figure 7). The DISCOVER study evaluated 820 surgeons for over 3 years; 29.2% of vitreoretinal surgeons reported intraoperative changes based on OCTi information [43].

The main recommendations for OCTi use are macular surgeries of the vitreoretinal interface for assessing ILM peeling, rhegmatogenous RDs for differentiating areas of retinoschisis and assessing tears, and tractional retinal detachment (TRDs) for assessing fibrovascular proliferation and identifying possible tears. Technologies under current development, such as gene therapy and retinal prosthesis implantation, will also benefit from the information provided by the OCTi [48].

The advancement of OCTi systems should facilitate intraoperative decision making in retinal surgeries. New perspectives in development involve software with real-time volumetric calculation, which is useful in gene therapy with the injection of subretinal medication, and adapted surgical instruments, which will reduce artifacts in the OCTi signal [49].

However, the excessive cost of OCTi systems remain an obstacle to its widespread use, especially in conjunction with the available platforms.

### 2.8. New Instrumentation in Vitreoretinal Surgery

Vitreoretinal surgery has considerably evolved in recent years, with progress occurring in vitrectomy probes (which tend to be smaller, faster, and safer) and all other aspects of the core instrumentation.

One of the first steps during vitrectomy is the insertion of scleral cannulas, and an auto inserter (Bausch + Lomb) that is used to automatically insert scleral cannulae instead of the conventional manual method is available. Automated insertion has significantly decreased the amount of pressure required to puncture a globe.

The cutter is one of the most important instruments in vitreoretinal surgery. This device allows for the effective removal of the hyaloid and vitreous base, core vitrectomy, and membrane cutting. Factors that have been modified to improve this instrument and reduce retinal traction are the blade design, duty cycle, cutting speed, and tip-to-port distance. Current vitreous cutters can deliver cut rates of up to 16,000 cpm depending on the vitrectomy platform (e.g., EVA) [50].

Another crucial step in vitrectomy surgery is the injection of vital dyes and perfluorocarbon, which is accomplished using a squeezer, a disposable device that consists of a silicone tube in a plastic frame; the squeezer has a Luer Lock for filling the silicon chamber and another to attach the silicon tip cannula. The Luer Lock prevents the backflow of the vitreous once the pressure is released.

Another modern instrument is the Sharkskin ILM forceps (Alcon), which uses a new technology that increases friction on the backside of the ILM forceps tip. The microstructured tip improves grasping, resulting in less shredding and less need for regrasping during peeling [51].

The new CryoPen (CryoTreq) (Vitreq, Vierpolders, The Netherlands) is a standalone, single-use, disposable cryopexy device that is about 20 cm long and a few centimeters thick without being connected to another device. Internally, for the required gas expansion, N_2_O patterns are used similarly to those used in the espuma devices often found in professional catering. A minimum of 15 to 20 cryopexy spots can be delivered over the lifetime of one disposable device. It is useful in cases of RD and retinopathy of prematurity [52].

### 2.9. Anti-VEGF Injection before PPV in Eyes with PDR and New Surgical Techniques

PPV often is indicated for persistent vitreous hemorrhaging, extensive fibrovascular proliferation threatening or involving the fovea, or TRDs with or without RRDs that occur in patients with PDR. The visual prognosis may be guarded in these patients because of the high incidence of intraoperative and postoperative complications [53]. Vitreous hemorrhages are the most common complication after PPV in patients with PDR, with incidence rates of up to 75% in some studies [54].

In eyes with advanced PDR characterized by ample, active neovascularization and/or extensive or multiple layers of fibrovascular proliferation, a preoperative intravitreal anti-VEGF injection may further decrease intraoperative hemorrhaging, facilitate fibrovascular membrane dissection [55,56], and reduce intraoperative and postoperative ocular complications [57].

Despite its proven efficacy in inhibiting neovascularization, intravitreal anti-VEGF in patients with PDR may induce fibrovascular contraction, leading to a TRD or aggravating a preexisting RD [58]. The progression of TRD typically manifests between 1 and 6 weeks following intravitreal anti-VEGF injection, with a mean onset of 13 days [59].

One study compared the 7-day and 20-day pre-vitrectomy administration of bevacizumab [60]. The clinical outcomes did not significantly differ between the two groups, but intraoperative severe bleeding, the frequency of the need for endodiathermy, iatrogenic retinal breaks, silicone oil tamponade, mean surgical time, and the rate of recurrent vitreous hemorrhages were increased in the 20-day group. However, the optimal interval time and intravitreal anti-VEGF doses need clarification.

Recent studies have shown that the perioperative application of anti-VEGF, before the end of vitrectomy, can also reduce intraoperative blood loss, reduce the level of VEGF, reduce the possibility of postsurgical complications, and improve ophthalmic parameters such as best corrected visual acuity, central macular thickness, and macular blood flow density [61].

### 2.10. Endoscopic Surgery

Endoscopy-assisted ophthalmic surgery is a relatively old technique, although it has not been widely adopted because of limitations such as the high cost of the instrument, steep learning curve, limited field of view, lack of stereopsis, and inability to perform bimanual procedures [62].

Endoscopy for posterior segment pathologies in patients with opaque corneas is a valuable diagnostic procedure. Since the introduction of the first described ophthalmic endoscope prototypes, the size, resolution, and maneuverability of these endoscopes have been optimized [63].

The use of an ophthalmic endoscope circumvents the limitations posed by poor visualization through the anterior segment [64], as in severe open-globe eye injuries that prevent surgeons from obtaining a clear view, and the anterior segment can be bypassed with the endoscope to achieve visualization. Studies comparing endoscopic vitrectomy to temporary keratoprosthesis for severe ocular trauma found that the surgical outcomes were similar, but patients were treated more quickly with endoscopic vitrectomy because this approach is less invasive and requires less preparation [65].

Another indication for endoscopic surgery is pediatric cases; the difficulty of working in the small vitreous cavity of a child increases the risk of iatrogenic lenticular trauma and retinal breaks [66]. Endoscopic vitrectomy facilitates sustained visualization, potentially reducing the risk of trauma [63].

Other indications for this technique include goniosynechialysis, retained lens fragments, a posteriorly dislocated intraocular lens, ciliary body photocoagulation, PVR, RD repair (especially for undetectable breaks in the peripheral retina), and endophthalmitis.

### 2.11. Management of Subretinal Hemorrhages

A subretinal hemorrhage can result from the presence of a neovascular membrane, ruptured arterial macroaneurysms, trauma, and the Valsalva maneuver. The toxic effects of subretinal blood can be demonstrated 24 h after hemorrhages, and the harmful retinal consequences can be attributed to the limited passage of nutrients, the shrinkage of the outer retinal layers due to clot formation, and the release of toxic substances such as iron, hemosiderin, and fibrin [67].

A variety of therapeutic options are available to treat subretinal hemorrhages, such as intravitreal gas administration, the intravitreal or subretinal application of recombinant tissue plasminogen activator (rt-PA) (Figure 8), the intravitreal injection of anti-vascular endothelial growth factor substances, and subretinal clot removal [68].

Several surgical techniques have been proposed to displace subretinal hemorrhages with variable success. There is no consensus on the optimal management of subretinal hemorrhages [69]. The intravitreal application of rt-PA with expansible gas is a less invasive procedure. A combination of PPV, subretinal rt-PA with 41-G retinotomy, and pneumatic displacement has been used with varying success rates.

### 2.12. Gene Therapy Using Viral Vectors

In ophthalmology, molecular genetics has substantially advanced and has been used to clarify diagnoses, direct counseling, and enable the first clinical trials of gene-based treatment in these diseases [70].

The first FDA-approved gene therapy in the United States is voretigene neparvovec-rzyl (Luxturna, Spark Therapeutics, Philadelphia, PA, USA) (Figure 9). This therapy, approved for treatment of biallelic *RPE65* inherited retinal dystrophy, uses an adeno-associated virus (AAV) 2-based vector that encodes the *RPE65* transgene [71]. Other gene therapies in development include NSR-REP1 (Nightstar Therapeutics, London, UK), an AAV-2-based therapy aimed at treating choroideremia, an X-linked recessive disease [72].

Current gene therapies use viral vectors to introduce a transgene into host cells. Alternative methods that do not require viral vectors, such as nanoparticles and iontophoresis, are being explored, but these are in varying stages of investigation.

### 2.13. Alternative Techniques for Refractory Macular Holes

A macula hole (MH) is a full-thickness retinal defect in the foveal center, with prevalence rates ranging from 0.02% to 0.8% in persons older than 40 years [73]. Although a high anatomic success rate can be achieved for idiopathic MHs (IMH) with PPV and ILM peeling (Figure 10), the treatment of large and/or refractory IMHs is a challenge [74].

A variety of modified surgical techniques to treat recurrent MHs have been described, including amniotic membrane grafting (Figure 10). In this technique, a human amniotic membrane patch placed under a MH helps to resorb subretinal fluid that may surround the recalcitrant MH, leading to improved visual acuity. Researchers have suggested that the amniotic membrane in the subretinal space serves as a scaffold for glial cell migration and enhances the adherence of the edges of the MH to the underlying RPE [75].

Another technique is the creation of a free ILM flap (Figure 11). The goal of the technique is to release ILM tension over the MH by creating a single continuous sheet of ILM that ends with a superior hinge beyond the MH; the ILM sheet then is draped back over the MH. This technique restores or maintains the integrity of the Müller cell footplates, which helps achieve a more physiologic postoperative foveal contour with less distortion [76].

Autologous retinal transplantation (ART) (Figure 12) may be a successful surgical option in patients with a refractory IMH. The theoretical advantage of ART is that the transplanted retina integrates into the adjacent tissue, potentially improving visual recovery compared with other inert tissue scaffolds [77].

The enlargement of ILM peeling, autologous platelet concentrate, lens capsule flap transplantation, and perifoveal hydrodissection are other surgical techniques described for refractory MHs. The surgical management of refractory MHs remains challenging and a controversial topic in vitreoretinal surgery [78].

Recent studies have shown that the rate of refractory IMH closure is similar between surgical techniques. In terms of visual recovery, the most efficient technique for treating those cases is amniotic membrane grafting, lens capsular flap transplantation, and autologous platelet concentrate, which allow for better functional results than a free ILM flap [78].

### 2.14. Perspectives for Stem Cell Therapy, RPE Transplantation, and Prevention of PVR

Retinal degenerative diseases such as age-related macular degeneration (AMD), retinitis pigmentosa, and Stargardt’s disease are characterized by the irreversible loss of RPE cells, photoreceptors, choriocapillaris, and other retinal cells [79]. The vision loss caused by these debilitating diseases has major impacts on mobility, independence, quality of life, and ability to function in the modern world.

Stem cell-based therapy is a potential approach to treat retinal degenerative diseases, and many animal studies and some clinical trials have reported encouraging results. When treating degenerative eye disorders, mesenchymal stem cells (MSCs) protect retinal ganglion cells and stimulate the regeneration of their axons in the optic nerve with the paracrine factors they secrete. MSCs provide a trophic supply of axonal neuroprotection and regeneration in damaged cells of the retina, either by the direct secretion of neurotrophic factors or by the stimulation of its endogenous cells, which provide additional paracrine supplies and/or effects of cell replacement when activated [80].

The results of subretinal pluripotent stem-derived RPE implantation (Figure 13) have been encouraging in preclinical models for AMD and Stargardt’s disease. The two main strategies for subretinal stem cell-derived RPE delivery are cell suspension implantation or sheets of cells on scaffolds [81].

Therefore, ongoing scientific experiments/research and activities to introduce MSCs into clinical practice offer great opportunities for cell-based therapy and highlight the essential role of MSCs in the evolution of ophthalmology.

Autologous RPE transplantation is a promising surgery [82]. The possibility of using the RPE from the same eye eliminates tissue rejection [83]; however, the secondary degeneration of photoreceptors and the risks of PVR induction using this technique result in longer follow-ups to determine the definitive benefits and evidence-based indications of the current technique [84].

The most important and serious cause of detachment surgery failure is PVR formation [85]. This disease results in re-detachments and risks of hypotony and globe atrophy. The risk factors of such a complication are the time from the RD to the final surgical intervention, genetic trends, residual vitreous after vitrectomy, preoperative and intraoperative bleeding and hypotony, the number and extension of retinal tears (including giant ones) allowing RPE cells to be in contact to the vitreous cavity, and retinal tears associated with trauma. Therefore, a personalized surgical technique for each case of RD and/or trauma including optimal vitreous removal, the control of intraoperative bleeding and/or hypotony, a low level of trauma during surgery, and the use of silicon oil in specific cases is an effective way to minimize this important complication [86,87]. Recently, groups of vitreoretinal surgeons worldwide have used intraoperative methotrexate at the end of surgery and during the postoperative period in repeated intravitreal injections without a specific evidence-based protocol. Therefore, randomized clinical trials are important in this field for better guidelines and the possible use of this adjunctive therapy to minimize PVR formation [88,89].

### 2.15. Port Delivery System

The Port Delivery System with ranibizumab (PDS) is a novel drug delivery device that is surgically implanted into the vitreous cavity and allows for the continuous release of the anti-VEGF ranibizumab. It eliminates the need for frequent intravitreal injections while maintaining therapeutic intraocular drug levels to control disease activity [90,91]. Ranibizumab formulation is customized and is different from the commercially available formulations for IVI. The customized preparation is stable under body temperature for a long time, leading to maintained drug delivery. The PDS was recently approved by the United States Food and Drug Administration for the treatment of neovascular age-related macular degeneration (nAMD) following the LADDER phase 2 and ARCH-WAY phase 3 clinical trials [92].

The implantation procedure for the PDS (Figure 14) is performed in the operation room under sterile conditions. Cautious Tenon’s capsule and conjunctival closure is a cardinal safety step to avoid conjunctival erosion, which is probably the most important factor in the prevention of endophthalmitis. Refilling is an intraoperative procedure that requires special preparation different from standard IVI including unique needles and the requirement of supplemental task lightening and magnification [92].

Endophthalmitis post-PDS has been found to occur at rates from 1.6% to 1.8%, which are significantly higher than the incidence of endophthalmitis associated with the IVI of anti-VEGF [90,93]. The majority of endophthalmitis cases are associated with conjunctival retraction. To decrease the possibility of endophthalmitis, a state-of-the-art surgical technique should be performed, emphasizing the necessity of avoiding traumatic conjunctival and Tenon’s capsule maneuvers and the adequate suture of such anatomical structures at the limbus that completely covers the PDS implant; additionally, care to avoid iatrogenesis and adverse ocular events including retinal detachment, retinal tear, choroidal detachment, and transient postoperative vision loss are key points [92].

Currently, the PDS has shown a promising efficacy and ability to mitigate treatment burden while effectively generating visual and anatomic outcomes similar to those in patients receiving the standard monthly ranibizumab for nAMD [90]. A recent study demonstrated an equivalent efficacy to monthly ranibizumab, with 98.4% of PDS-treated patients not receiving supplemental treatment in the first 24-week interval [90]. Further studies investigating this novel drug delivery system in other disease states are ongoing [91].

## 3. Conclusions

The main objective of this review is to highlight the recent advances in vitreoretinal surgery. Vitreoretinal surgery has become safer and more effective due to advances in devices, techniques, and treatments using stem cells and gene therapy. Surgeons should be updated on advances in eye surgery to guide patients towards the best treatments, refer patients to specialists to treat diseases considered untreatable in the past, and perform personalize surgeries based on robust scientific levels of evidence of safety and efficacy.

## Figures and Tables

**Figure 1 jcm-11-06428-f001:**
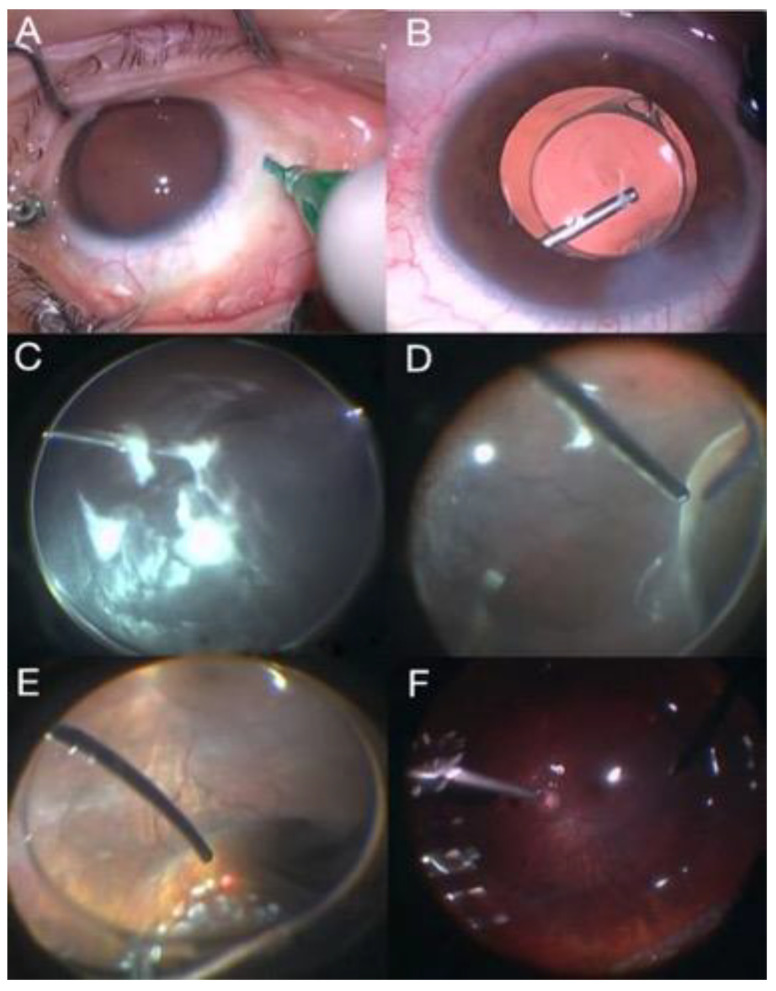
Primary vitrectomy surgery: (**A**) sclerotomies, (**B**) intraocular lens implantation, (**C**) intravitreous triamcinolone injection, (**D**) vitreous base removal, (**E**) laser application in retinal breaks, and (**F**) fluid–air exchange.

**Figure 2 jcm-11-06428-f002:**
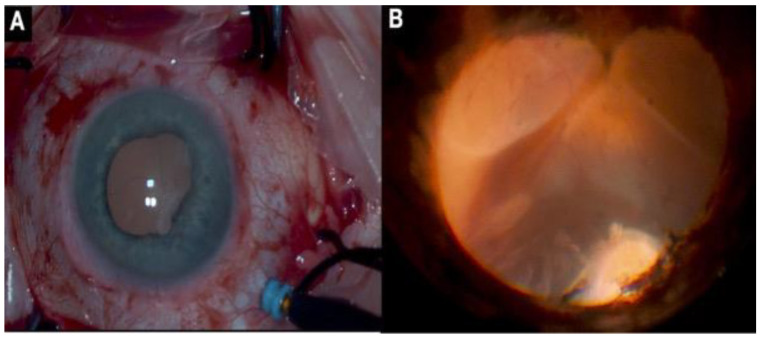
(**A**) Scleral buckling with chandelier illumination; (**B**) transscleral cryotherapy in the region of retinal breaks.

**Figure 3 jcm-11-06428-f003:**
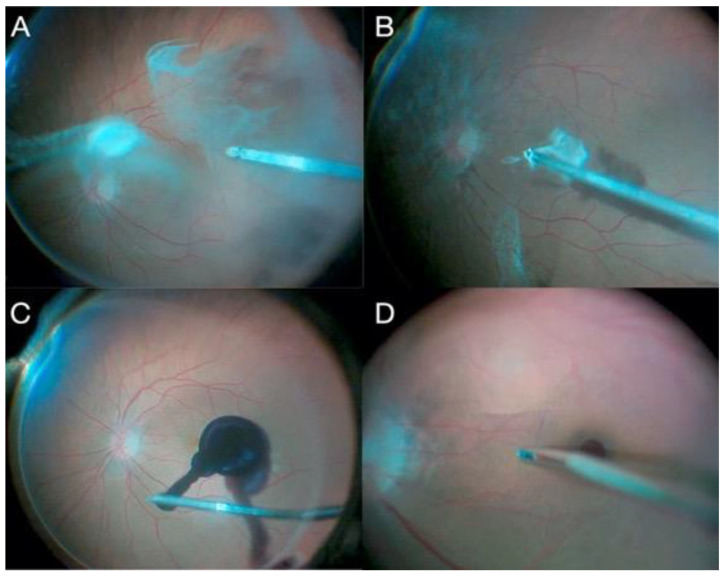
Conventional MH vitrectomy surgery: (**A**) triamcinolone injection, (**B**) posterior hyaloid detachment, (**C**) brilliant blue injection, and (**D**) ILM peeling.

**Figure 4 jcm-11-06428-f004:**
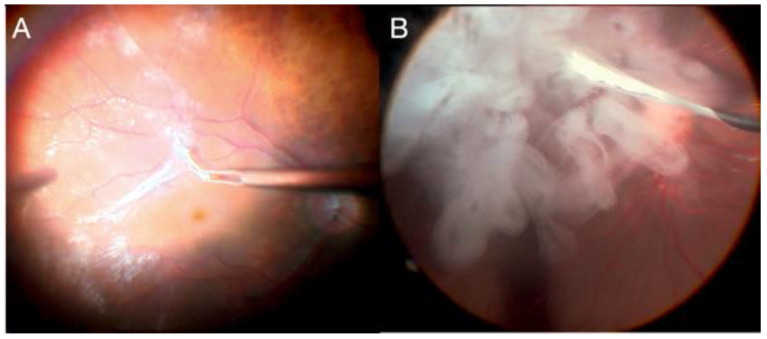
(**A**) ERM peeling; (**B**) triamcinolone injection for vitreous identification.

**Figure 5 jcm-11-06428-f005:**
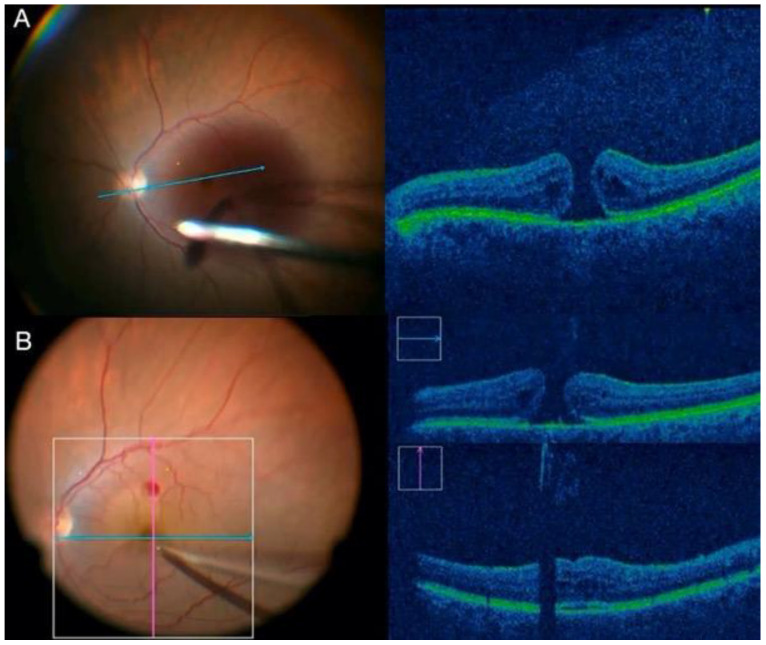
(**A**) injection of dye extracted from açaí (phase I/II of clinical trial); (**B**) ILM peeling.

**Figure 6 jcm-11-06428-f006:**
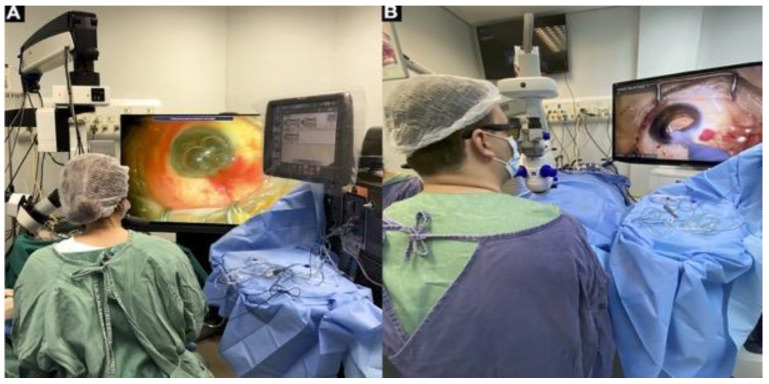
The 3D heads-up visualization system: (**A**) Alcon Ngenuity; (**B**) Carl Zeiss Artevo 800.

**Figure 7 jcm-11-06428-f007:**
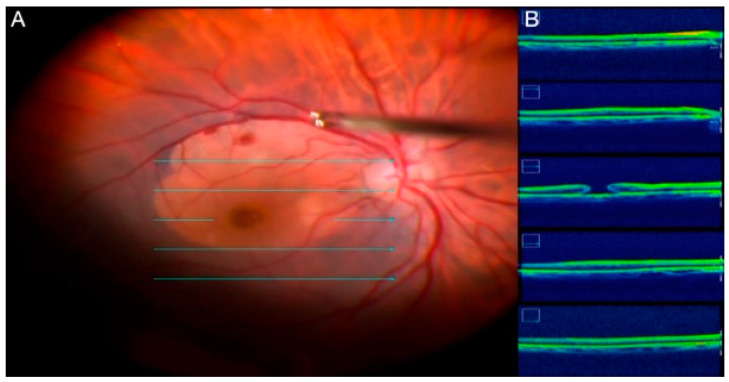
(**A**) ILM peeling; (**B**) intraoperative image of posterior vitrectomy guided by real-time OCTi.

**Figure 8 jcm-11-06428-f008:**
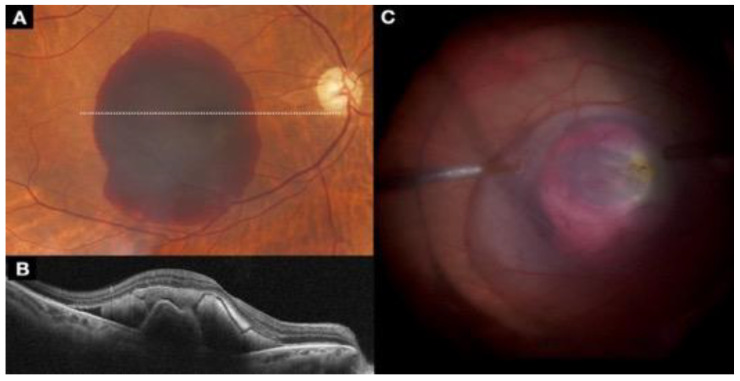
(**A**) The fundus image shows a subretinal hemorrhage; (**B**) cross-section spectral-domain OCT shows subretinal hemorrhage and a pigment epithelium detachment; (**C**) intraoperative image of PPV with subretinal application of rt-TPA to manage the subretinal hemorrhage.

**Figure 9 jcm-11-06428-f009:**
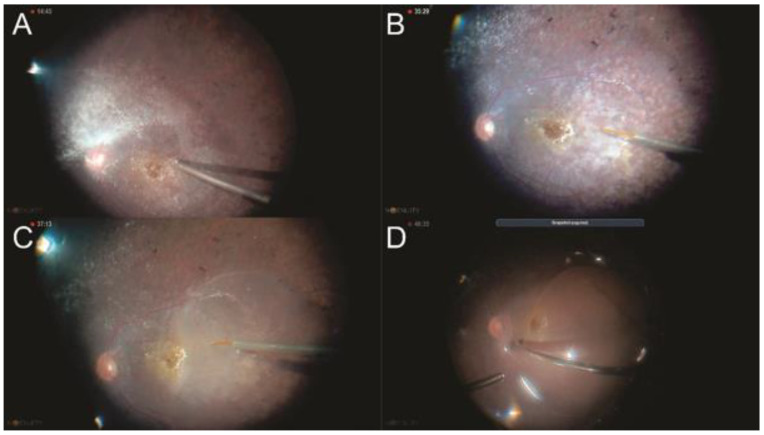
(**A**) Posterior hyaloid detachment; (**B**) 41-gauge canula penetration at subretinal space; (**C**) subretinal voretigene neparvovec injection; (**D**) fluid–air exchange.

**Figure 10 jcm-11-06428-f010:**
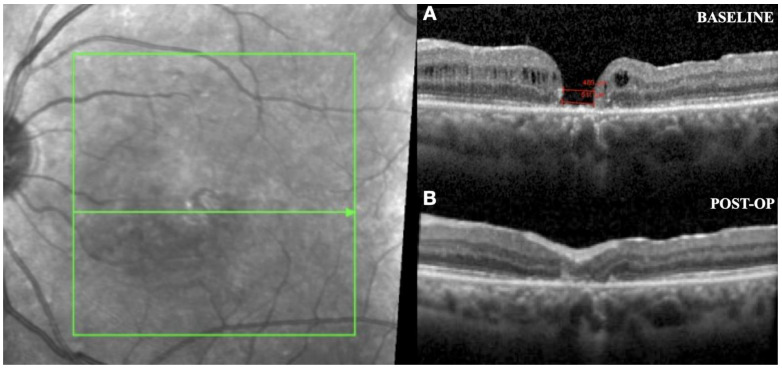
(**A**) OCT image shows a preoperative MH; (**B**) postoperative outcome of MH closure surgery with an amniotic membrane graft.

**Figure 11 jcm-11-06428-f011:**
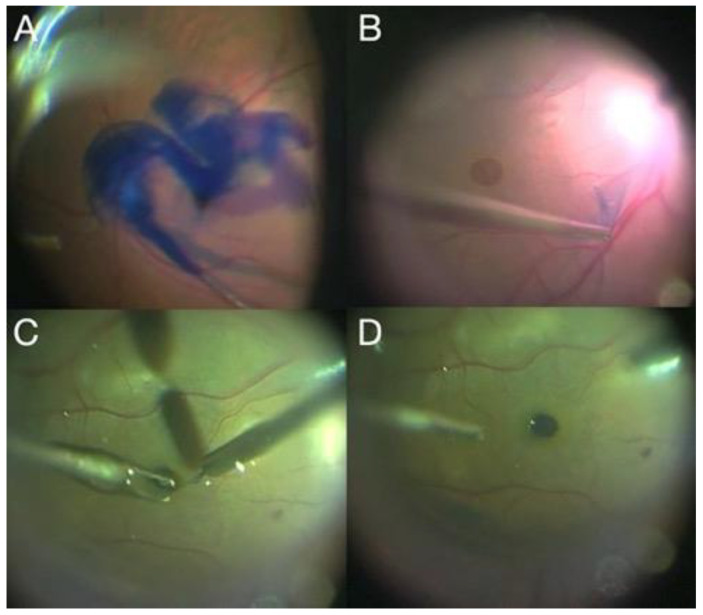
(**A**) Brilliant blue injection; (**B**) ILM peeling; (**C**) ILM transplant placement; (**D**) final result of intraoperative ILM transplantation.

**Figure 12 jcm-11-06428-f012:**
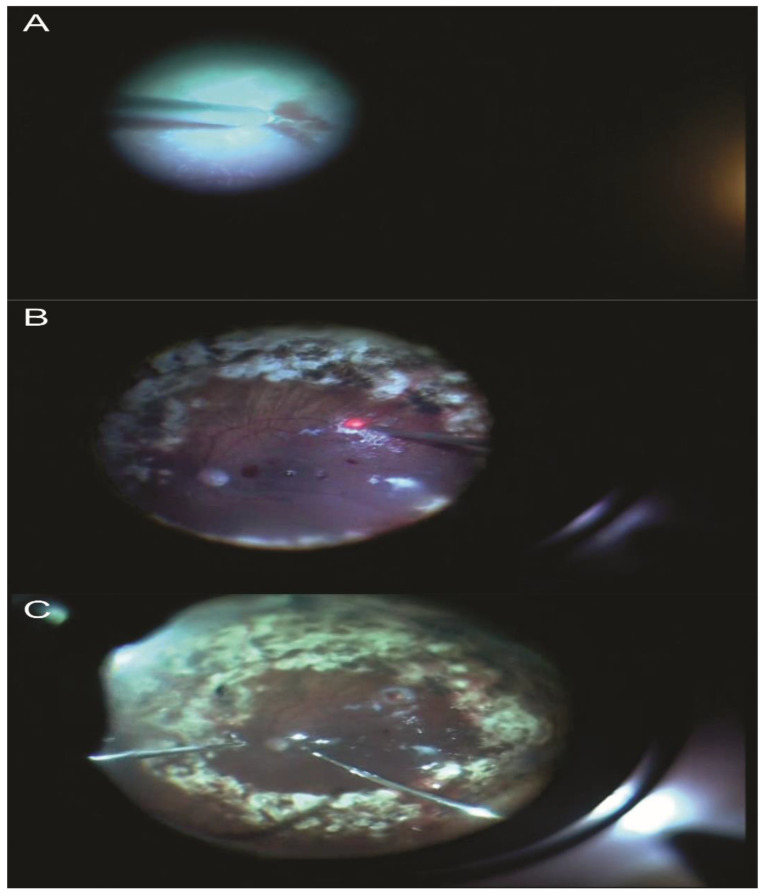
(**A**) Full-thickness autologous retinal graft taken from the temporal superior arcade area; (**B**) laser application in the area of graft removal; (**C**) retinal autologous transplant placement.

**Figure 13 jcm-11-06428-f013:**
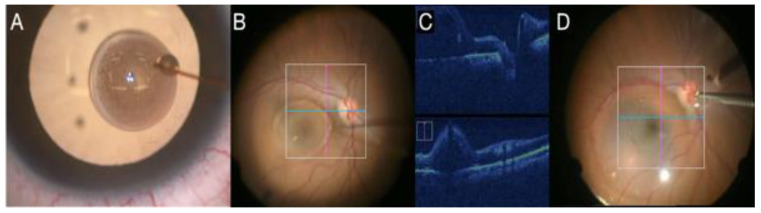
(**A**) Stem cells over the cornea; (**B**) subretinal injection of solution for stem cell therapy (phase I of clinical trial); (**C**) intraoperative image of subretinal stem cell injection by real-time OCTi; (**D**) final result after subretinal stem cell injection.

**Figure 14 jcm-11-06428-f014:**
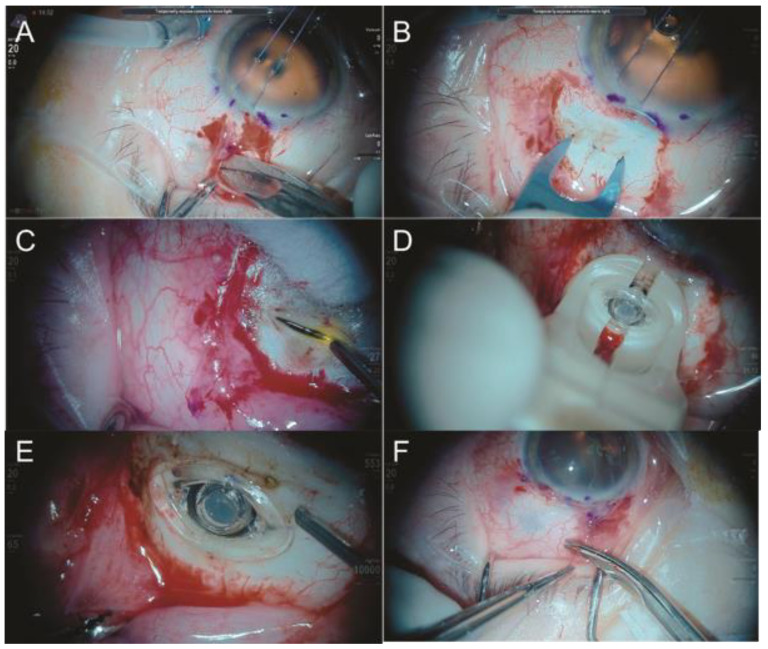
(**A**) Opening the conjunctiva and tenon using non-traumatic surgical technique; (**B**) marking of the correct PDS implantation points, as well as a scleral incision of 3.5 mm, until the visualization of the choroid; (**C**) external laser at the choroid to avoid bleeding; (**D**) PDS implantation; (**E**) vitrectomy of the vitreous around the PDS implant; (**F**) suture of both conjunctiva and tenon at the limbus and covering the PDS implant using no traumatic technique.

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
