# Peer review of "Advances in Vitreoretinal Surgery"

_jcm, 2022, doi:10.3390/jcm11216428_

Round 1

Reviewer 1 Report

This review article provided us the history and up-to-date development of vitrectomy and its related technology, which now been widely used in our routine clinics and also the direction of future development. However, the logical construction of this article make readers a little confuses, authors combined the development of single technology or instrument with the retinal diseases such as RRD, and subretinal hemorrhage, I would suggest using the state line of the advances of Vitreoretinal Surgery, from the choice of surgery (from scleral buckling to PPV), Vitreoretinal Surgery related instruments (from operation system to instruments, staining, et al), perioperative period drug using( including anti-VEGF, tPA), new methods (from gene therapy to stem cell) , and future direction.

There are a few issues/questions that need more clarity:
1. In 2.1.2 scleral buckling section, another method using surgical microscope for sclera buckling without a chandelier lighting system was also use by some retinal surgery (PMID: 24790997; PMID: 21323267), this method also very applicable.

2. authors should add the development of intraocular tamponades, such as BSS, gas, and oil.

3. 2.8. Bevacizumab before Vitrectomy in Eyes……section, it may change to “anti-VEGF injection  perioperative period”, since there have been many anti-VEGF drugs available. Authors should refer to more recent articles about the use of anti-VEGF before vitrectomy, such as the time, type, pre-surgical, during surgical, after surgical.

4. authors may add a new section, about the benefit from the advances of the vitreoretinal surgery, in this section, authors could list the retinal diseases, such as the change of the operation opportunity (like the vitreous hemorrhage in PDR), the vision gain in macular hole,.

Reviewer 2 Report

Dear authors,

Well written and useful article.

I add some comments:

When talking about chandelier in scleral buckling also mention the 27G chandelier.

I would also mention that using smaller gauges sclerotomies the need of suturing is also reduced, this causes less conjunctival inflammation and less discomfort for patients.

Chromovitrectomy: mention MembraneBlue-dual (R) that stains ILM and ERM

Author Response

Por favor, verifique o anexo.

Round 2

Reviewer 1 Report

the article has improved much after revising